# The Role of Gut Microbiome in the Pathogenesis of Multiple Sclerosis and Related Disorders

**DOI:** 10.3390/cells12131760

**Published:** 2023-06-30

**Authors:** Anna Dunalska, Kamila Saramak, Natalia Szejko

**Affiliations:** 1Department of Neurology, Medical University of Warsaw, 02-097 Warsaw, Poland; adunalska@gmail.com; 2Department of Neurology, Hochzirl Hospital, 6170 Hochzirl, Austria; kamilasaramak@gmail.com; 3Department of Clinical Neurosciences, University of Calgary, Calgary, AB T2N 1N4, Canada; 4Department of Bioethics, Medical University of Warsaw, 02-091 Warsaw, Poland

**Keywords:** multiple sclerosis, brain-gut axis, gut microbiome, neuromyelitis spectrum disorders

## Abstract

Multiple sclerosis (MS) is a chronic, progressive neuroinflammatory disease with a complex pathophysiological background. A variety of diverse factors have been attributed to the propagation of inflammation and neurodegeneration in MS, mainly genetic, immunological, and environmental factors such as vitamin D deficiency, infections, or hormonal disbalance. Recently, the importance of the gut-brain axis for the development of many neurological conditions, including stroke, movement disorders, and neuroinflammatory disorders, has been postulated. The purpose of our paper was to summarize current evidence confirming the role of the gut microbiome in the pathophysiology of MS and related disorders, such as neuromyelitis optica spectrum disorder (NMO-SD). For this aim, we conducted a systematic review of the literature listed in the following databases: Medline, Pubmed, and Scopus, and were able to identify several studies demonstrating the involvement of the gut microbiome in the pathophysiology of MS and NMO-SD. It seems that the most relevant bacteria for the pathophysiology of MS are those belonging to *Pseudomonas*, *Mycoplasma*, *Haemophilus*, *Blautia, Dorea*, *Faecalibacterium*, *Methanobrevibacter*, *Akkermansia*, and *Desulfovibrionaceae genera*, while *Clostridium perfringens* and *Streptoccocus* have been demonstrated to play a role in the pathophysiology of NMO-SD. Following this line of evidence, there is also some preliminary data supporting the use of probiotics or other agents affecting the microbiome that could potentially have a beneficial effect on MS/NMO-SD symptoms and prognosis. The topic of the gut microbiome in the pathophysiology of MS is therefore relevant since it could be used as a biomarker of disease development and progression as well as a potential disease-modifying therapy.

## 1. Introduction

Multiple sclerosis (MS) is a progressive neuroinflammatory and neurodegenerative disease that usually has an onset in early adulthood and is predominantly found in females [1]. The main pathophysiological mechanism behind the development of MS is the activation of immunity, which was originally attributed to a T-mediated response, although, more recently, lymphocytes B and microglia have also been found to be involved [2]. As for other factors that have an influence on the initiation and propagation of this process, the importance of such causes as genetic contribution [3], low levels of vitamin D [4], infections, especially a history of Epstein–Barr virus exposure [5], smoking [6], and obesity [7], among others, has been shown. One of the risk factors that has been extensively investigated in the last few years, not only in the context of demyelinating disorders but also a variety of other neurological conditions, is the influence of the gut microbiome and microbiota [8]. While the microbiome includes both the host environment and the variety of microorganisms populating it, the term “microbiota” is used to define only microbiota specific to a particular host or disease [9]. The so-called gut-brain axis [10] is a term used to denote the interaction between the microbiome and the brain. It has been demonstrated that changes in the gut microbiome cause inflammation that affects a variety of systems [11,12], including the central and peripheral nervous systems (CNS) [13,14,15,16]. These interactions have been shown to take place at different levels and involve influences on brain signaling [17], gut-projecting spinal afferent neurons [18], gut-projecting afferent neurons [19], the influence of gut microbiota on the immune system [20], and metabolic systems [21], which, in turn, have an impact on the central and peripheral nervous systems. In the recently published systematic review by Tilocca et al. [22], the authors summarized previous evidence regarding different multi-omics techniques such as metagenomics, metatranscriptomics, metaproteomics, and metabolomics used for the investigation of gut-brain axis interactions. We were able to identify several key components of the gut-brain axis: the CNS, the autonomic nervous system, the enteric nervous system, the hypothalamic-pituitary-adrenal axis, and the immune system.

Especially the impact on the immune system has been postulated as the main mechanism of microbiome influence on the development of MS [23]. However, it is still not clear whether changes in the gut microbiome occur prior to the disease onset in the prodromal stage and could have a causal impact on the disease initiation, whether they are a type of biomarker of the disease progression and occur when the disease propagates, or whether they are a direct consequence of the disease-modifying therapies. It is also possible that multiple mechanisms and bidirectional influences have an impact on the gut-brain axis in MS (Figure 1). One of the suggested mechanisms is activation by the infection [24] either viral [25,26,27], bacterial [27], or fungal [28], that in turn leads to a number of further changes, such as activation of the immune system and changes in the microbiome composition [29]. Another useful approach in this context is offered by metagenomics [30], which comprises both microbiological as well as genetic techniques. As a result, metagenomics enables the identification of DNA specific to particular microorganisms [31].

The purpose of our paper was to summarize current evidence behind the role of the gut microbiome in the pathophysiology of MS and related disorders. We are also mentioning the potential use of agents that interfere with the composition of the gut microbiome, such as probiotics and fecal transplantation, for the therapy of MS. To identify articles eligible for inclusion, we have searched electronic databases (Pubmed, Medline, and Scopus) with the main focus on original research. We did not use any time restrictions to include the articles.

## 2. The Role of Gut Microbiome in the Pathogenesis of Multiple Sclerosis

Taking into consideration the fact that the gut microbiota is crucial for the immune system’s growth and maturation [32], it is intuitive to expect its contribution to the pathogenesis of MS [33,34,35].

Animal models provide information that MS-linked microbiota produce factors that precipitate an MS-like autoimmune disease [36]. Molecular mimicry between gut bacterial components and central nervous system (CNS) autoantigens, in concert with gut microbes inducing Th17 cells, acts together to worsen CNS autoimmunity in experimental autoimmune encephalomyelitis (EAE), the most widely used animal model of MS [37]. Therefore, the main mechanism behind the creation of this animal model is based on gut-brain interaction. In the process of developing this model, it has been found that antibiotic administration protected against the disease, which was attributed to Treg- and Th2-cell responses [38].

Following these initial findings, immune cells from mice that received microbiota samples from MS individuals produced less interleukin (IL)-10 than immune cells from mice colonized with healthy samples. This is of particular importance since IL-10 may have a regulatory role in spontaneous CNS autoimmunity [39]. Increased *Streptococcus* concentration, a higher *Firmicutes*/*Bacteroidetes* ratio, and decreased *Prevotella* concentration are associated with higher disease activity and more abundant intestinal Th17 cells [37]. Proinflammatory responses in human peripheral blood mononuclear cells and in monocolonized mice are related to *Akkermansia muciniphila* and *Acinetobacter calcoaceticus* colonization, which are typical for MS patients [40]. On the other hand, *Parabacteroides distasonis*, which is reduced in MS patients, stimulates anti-inflammatory IL-10 expressing human CD4+CD25+ T cells and IL-10+FoxP3+ Tregs in mice [40]. Microbiota transplantation from MS patients into germ-free mice results in more severe manifestations of EAE compared with mice ‘humanized’ with microbiota from healthy controls (HC) [40]. Moreover, dysbiosis can modulate immunological responses to the microbiota and affect the integrity of the epithelia that comprise cellular barriers vital for the integrity of the intestine and CNS [41]. In the EAE animal model, increased intestinal permeability, overexpression of the tight junction protein zonulin, alterations in intestinal morphology, increased infiltration of proinflammatory Th1/Th17 cells, and a reduced regulatory T cell number in the gut lamina propria, Peyer’s patches, and mesenteric lymph nodes were observed [42]. Germ-free mice are also reported to have greater permeability of the blood-brain barrier (BBB) compared to pathogen-free mice with a normal gut flora [43]. The low-grade microbial translocation to the systemic circulation and ultimately to the brain is a crucial point regarding the contribution of intestinal permeability changes to MS pathophysiology [44]. The change in gut microbiome achieved by exposing germ-free adult mice to a pathogen-free gut microbiota decreased BBB permeability and up-regulated the expression of tight junction proteins [43]. The same study delivered information about the presence of hypermyelinated axons within the prefrontal cortex of germ-free mice, suggesting the role of the microbiota in controlling myelin production in this brain area [43]. In another study by Colpitts et al. [45], they compared gut microbiota composition between acute inflammatory and chronic progressive forms in a murine model of secondary-progressive multiple sclerosis. As a result, the authors found that the mice that developed a severe secondary form of EAE exhibited a dysbiotic gut microbiome when compared with the healthy control mice. The authors also complemented this study with a sub-analysis regarding the influence of antibiotic therapy on the outcomes of the progressive stage of EAE. Interestingly enough, mice receiving antibiotics demonstrated reduced mortality and disease severity.

According to the literature based on studies in humans, specific bacterial taxa are significantly associated with the pathophysiology of MS. In particular, the following bacteria were found to be more abundant in MS patients than controls: *Pseudomonas*, *Mycoplana*, *Haemophilus*, *Blautia*, *Dorea*, *Faecalibacterium*, *Methanobrevibacter*, *Akkermansia*, and *Desulfovibrionaceae genera* [46,47,48,49]. These taxa have been found to be associated with variations in the expression of genes involved in dendritic cell maturation, interferon signaling, and NF-kB signaling pathways in circulating T cells and monocytes. It has also been postulated that *Akkermansia*, found more abundantly in MS patients than healthy controls, may be a compensatory response in MS. What is more, progressive MS is uniquely linked to elevated *Enterobacteriaceae* and *Clostridium g24 FCEY* and decreased *Blautia* and *Agathobaculum* colonization [39]. Moreover, Clostridial species associated with MS might be distinct from those broadly associated with other autoimmune conditions [50]. Differences in the microbiome composition between MS patients and healthy controls concern clostridial species belonging to *Clostridia* clusters XIVa and IV and *Bacteroidetes* [50]. As clusters IV and XIVa of the genus *Clostridium* promote Treg cell accumulation and the colonization of mice by a defined mix of Clostridium strains is described to provide an environment rich in transforming growth factor–b and affect Foxp3+ Treg numbers, it can alter colon function and disrupt immune homeostasis [51]. In addition, several Clostridium species were associated with higher disability scores measured with the expanded disability status scale (EDSS) and fatigue scores [39].

Another important topic is the influence of disease-modifying therapies on the microbiome’s composition. MS patients on disease-modifying therapies have been found to have increased fecal concentrations of *Prevotella*, *Sutterella*, and *Akkermansia* and decreased *Sarcina* in comparison to untreated individuals [49,52,53]. Glatiramer acetate treatment is described as being linked to differences in microbiome composition in MS patients, expressed in abundance of *Bacteroidaceae*, *Faecalibacterium*, *Ruminococcus*, *Lactobacillaceae*, *Clostridium*, and other *Clostridiales* [49]. On the other hand, in untreated MS patients, an increase in the *Akkermansia, Faecalibacterium*, and *Coprococcus genera* after vitamin D supplementation was observed [49].

Studies also show that manipulations of the gut microbiome through the use of probiotics have a positive effect on the health of patients with MS [54,55,56,57]. Administration of probiotics increases the colonization of some taxa known to be depleted in MS, such as *Lactobacillus*, and reduces the abundance of *Akkermansia* and *Blautia*, linked to dysbiosis in MS [56]. Methane metabolism, number of inflammatory monocytes, mean fluorescence intensity (MFI) of CD80 of classical monocytes, and HLA-DR MFI on dendritic cells are also reduced after probiotic intake in MS patients [55]. Moreover, in a healthy population, probiotic administration is linked to decreased expression of the MS risk alleles HLA-DQA1 and HLA-DPB1 [56]. Administration of *Lactobacillus acidophilus*, *Lactobacillus casei*, *Bifidobacterium bifidum*, and *Lactobacillus fermentum* has favorable effects on disability levels assessed with EDSS, parameters of mental health, inflammatory factors, markers of insulin resistance, and HDL- and total-/HDL-cholesterol levels in MS patients [57]. Probiotic supplementation down-regulates gene expression of IL-8 and tumor necrosis factor–alpha (TNF-a) mRNA in peripheral blood mononuclear cells of patients with MS [57]. It has also been observed that the low-grade inflammation linked to chronic inflammatory diseases is largely combated by nutrients, including nondigestible dietary fibers [58]. Their action is mediated by the gut microbiota, and any microbial change brought on by diet alters host-microbe interactions in a way that either ameliorates or exacerbates the disease, which also applies to MS [57]. What is more, intermittent fasting (IF) ameliorates the clinical course and neurodegenerative changes in EAE [59,60,61]. IF also increased the diversity of gut bacteria, particularly those belonging to the families of *Lactobacillaceae*, *Bacteroidaceae*, and *Prevotellaceae*. IF reduces the IL-17-producing T cells in the gut while increasing the regulatory T cells [59,60,61].

The overview of articles discussed in this section is presented in Table 1.

## 3. The Role of Gut Microbiome in the Pathogenesis of Neuromyelitis Optica Spectrum Disorders

Neuromyelitis optica spectrum disorders (NMO-SDs) are a group of chronic autoimmune diseases of the CNS [62]. The symptoms of NMO-SDs are caused by demyelinating lesions occurring mainly in the spinal cord and optic nerves [63,64]. The majority of patients are seropositive for autoantibodies (NMO-SD IgG) against aquaporin-4 (AQP4), a water channel expressed in astrocytes. The AQP4-specific autoantibodies are classified as immunoglobulin (Ig)G1, a T cell-dependent Ig subclass [63]. Although great progress has been made to unravel the pathogenesis of NMO-SDs, the environmental triggers underlying the production of NMO-SD IgG remain unclear [63,65].

To date, research has shown that T cells recognizing the epitope of AQP4 display cross-reactivity to homologous peptide sequences of commensal bacteria found in human gut flora [66]. In 2012, Varrin-Doyer et al. conducted the first study using peripheral blood T cells obtained from NMO-SD patients and healthy controls (HC) [66]. Both T cells from NMO-SD patients and HC proliferated to intact AQP or discrete AQP4 peptides. The T cells from NMO-SD patients *showed* markedly higher *proliferation*, especially when exposed to the peptide p61-80. The peptide p63-76, in turn, exhibited strong homology to a sequence within the *Clostridium perfringens* adenosine triposphate-binding cassette (ABC) transporter permease. The T cells from NMO-SD patients also proliferated to this homologous bacterial sequence, showing cross-reactivity and supporting the role of molecular mimicry. Moreover, monocytes from NMO-SD patients produced a greater amount of interleukin (IL)-6, which is responsible for T17 polarization in this group of patients [66]. The paper published by Cree et al. also supports the role of *C. perfringens* in the pathogenesis of NMO-SDs [67]. The stool samples obtained from 16 NMO-SD patients showed a significantly higher concentration of several microbial communities, especially *C. perfringens*, in comparison to the ones obtained from 16 HC. Most of the NMO-SD patients were treated with immunotherapy, mostly rituximab, which could also influence the gut microbiome. In addition, the authors compared these findings with those of 16 MS patients, five of whom were treated with rituximab. Notably, the gut microbiota in NMO-SD and MS patients differed significantly despite treatment with rituximab in both groups [67]. On the other hand, the study conducted by Gong et al. in China revealed the overrepresentation of *Streptococcus* sp. in the fecal microbial composition of the NMO-SD patients [68]. Interestingly, the abundance of *Streptococcus* sp. was positively correlated with disease severity, and the use of immune suppressant medications had a depleting influence on the gut microbiome. Moreover, the patients with NMO-SD showed significant reductions in faecal butyrate, which is believed to have an anti-inflammatory effect [68]. The anti-inflammatory effect of short-chain fatty acids is not limited to the intestinal tract; it also increases the Treg level and inhibits Th17 cell differentiation [68]. The same group from China showed that although in the sigmoid mucosal biopsies collected from 6 NMO-SD patients the diversity of bacterial flora was overall diminished, *Streptococcus* and *Granulicatella* sp. were still amply detected in the samples [69]. Furthermore, through the decreased expression of tight junction proteins, the integrity of the intestinal barrier may be impaired. Additionally, the increased number of plasma cells, macrophages, and mast cells found in the lamina propria suggests inflammatory activation of the gut in patients with NMO-SD. Another study from China supports the aforementioned results, showing an overrepresentation of pathogenic species such as *Streptococcus* and *Flavonifractors* in the stool samples of NMO-SD patients [70]. Zhang et al. characterized the gut microbiota in both AQP4 seropositive and AQP4 seronegative groups of NMO-SD patients separately and compared these findings with the HC [71]. The microbial composition in NMO-SD showed an increased prevalence of *Proteobacteria*, *Bacteroidetes,* and *Firmicutes*, respectively. Furthermore, butyrate-producing species were abundantly represented in HC compared to NMO-SD patients [71]. Pandit et al. investigated the blood and stool samples of 39 Indian patients with NMO-SD [72]. The prevalence of *Clostridium boltae* was significantly higher in the stool samples obtained from AQP4 positive patients compared to seronegative specimens. *C boltae* was not detected in the stool samples collected from HC. Moreover, *C boltae* peptide p 59–71 showed homology with AQP peptide p92–104. The presence of *C boltae* correlated with the expression of inflammatory genes associated with B cell chemotaxis as well as Th17 cell activation [72]. Recently, Cheng et al. investigated the role of T follicular helper (Tfh) cells in NMO-SD recurrence and evaluated whether the levels of glycoursodeoxycholic acid (GUDCA), a microbiota metabolite, influenced levels of serum C-X-C motif ligand 13 (CXCL13), which reflect the effects of the Tfh cells on B-cell-mediated humoral immunity [73]. The level of GUDCA was higher in patients with NMOSD with low activity, which was positively correlated with CXCL13. The overview of studies presented in this section is presented in Table 2.

## 4. Conclusions

The current degree of evidence supports the importance of the gut microbiome in the pathophysiology of MS and related disorders. On the one hand, there are many taxa of bacteria that are found more frequently in patients with MS, but also, disease-modifying therapies used in MS have been shown to change the microbiome composition. One important point that should be considered is whether these changes occur prior to the occurrence of MS or are secondary to the disease itself [74]. It is also possible that genetically determined microbiome changes determine the microbiome composition, which, in turn, leads to disease progression [75,76,77]. Another issue is related to the differences between the microbiome, which constitutes both the host environment as well as all microorganisms encountered in it, and the microbiota, which is limited to microorganisms only [78]. In addition, it is still not clear whether the totality of the patient’s microbiome is equally important as microbiomes in different locations, such as the gut, oral [79], nasal cavity [80], pulmonary tract [81], or vagina [82]. This is of great importance since the environment is the single most important determinant of microbiota composition [83]. Another topic that still needs exploration is the influence of different environmental factors on the microbiome composition and its interaction with MS pathophysiology. Previous studies have demonstrated that dietary changes [84], stressors [85], substance abuse [86], and chronic illnesses [87] co-existing with MS and pharmacotherapy, especially antibiotics [88], influence the microbiome composition.

There are several limitations to our review: (i) although we used a detailed search strategy, it is still possible that we did not include some important studies, especially the ones posterior to publication of our review; (ii) another limitation is related to the inclusion of articles that are written only in English; (iii) finally, we mainly focused on the microbiome in our review; however, it is highly dependent on environmental factors [89], especially diet, which also plays an important role in the development of MS [90].

It can therefore be concluded that although there are some preliminary data suggesting that the gut microbiome plays an important role in the pathophysiology of MS, the microbiome composition is determined by too many confounding factors. Future studies should be focused on overcoming this limitation, mainly using methodologies derived from population genetics, such as Mendelian randomization [91].

## Figures and Tables

**Figure 1 cells-12-01760-f001:**
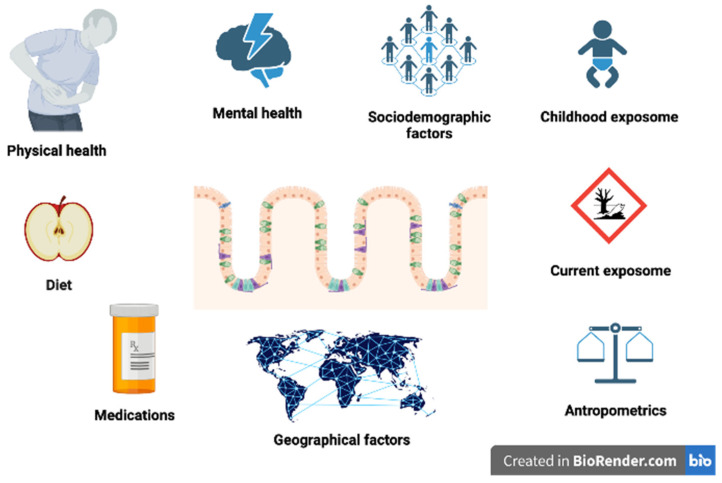
Various environmental factors that have influence on gut microbiome.

**Table 1 cells-12-01760-t001:** An overview of studies investigating the role of gut microbiome in the pathogenesis of multiple sclerosis. Studies are presented in chronological order.

Reference	Number of MS Participants	Methodology	Results
Animal models			
Nouri et al. [42]	NA	Analysis of intestinal permeability in EAE mice	Increased intestinal permeability, overexpression of the tight junction protein zonulin and alterations in intestinal morphology in EAE Adoptive transfer to healthy mice of T cells, isolated from EAE-diseased animals, led to intestinal changes similar to those resulting from the immunization procedure
Braniste et al. [43]	NA	Assessment of the importance of the intestinal microbiota in the maintenance of BBB integrity in a mouse model. Assessment whether these changes are present during different stages of life.	Germ-free mice displayed increased BBB permeability compared to mice with a normal gut flora. Exposure of germ-free adult mice to a pathogen-free gut microbiota decreased BBB permeability and up-regulated the expression of tight junction proteins. These changes were found to have initiated during gestation and propagated throughout life.
Cekanaviciute et al. [40]	36 (for animal model analysis)	Combination of animal model and clinical research Comparison of GM in MS patients and HC Microbiota transplants from MS patients to germ-freem mice	Gut microbiota from MS patients induced more pronounced EAE in mice
Colpitts et al. [45]	45	Comparison of GM in EAE and HC Comparison of clinical course after antibiotics administration	EAE mice developed dysbiosis Better prognosis after antibiotic therapy
Berer et al. [52]	47 (for animal model analysis)	Transplantation of GM from MS patient to mice with EAE Evaluation of immunological, GM changes in the mice after transplantation	MS derived microbiota induced a significantly higher incidence of autoimmunity The microbial profiles of the colonized mice showed a high intraindividual and temporal stability Immune cells from mouse recipients of MS samples produced less IL-10
Miyauchi et al. [37]	14	3. Change of GM with antibiotics 4. Analysis of MOG-specific response in the gut	4. Two signals from GM activate autoreactive T cells in the small intestine that respond specifically to MOG 5. After EAE induction, MOG-specific CD4 T cells were detected in the small intestine. 6. *Erysipelotrichaceae* and *Lactobacillus reuteri* induced EAE in mice model
Studies in humans			
Miyake et al. [50]	20	Comparison of fecal symptoms between MS and HC	Significant differences between MS and HC when it comes to the following bacterial taxa: clostridial species belonging to *Clostridia* clusters XIVa and IV and Bacteroidetes.
Cantarel et al. [49]	7	Comparison of GM in MS and HC Evaluation of the influence of GA and Vit. D treatment on the GM	The abundance of *Faecalibacterium* was lower in MS GA therapy leaded to changes in GM composition in MS subjects Vit. D supplementation leaded to increase in the *Akkermansia*, *Faecalibacterium*, and *Coprococcus* in MS patients
Tremlett et al. [47]	18	GM comparison between MS and HC children	MS cases had a significant enrichment in the *Desulfovibrionaceae* (*Bilophila*, *Desulfovibrio* and *Christensenellaceae*) and depletion in *Lachnospiraceae* and *Ruminococcaceae*
Jangi et al. [48]	60	16S rRNA sequencing to compare GM in MS and HC Correlation of GM changes with gene expression Comparison of proliferation and cytokine assays in response to specific microbial stimulation. Sera from MS and HC was collected for ELISA-based techniques to capture serologic activity directed against specific GM. Breath samples from MS and HC were collected to determine breath methane concentrations. Influence of DMT on GM in MS	GM alterations in MS included increases in *Methanobrevibacter* and *Akkermansia* and decreases in *Butyricimonas* Changes in GM correlated with variations in the expression of genes involved in dendritic cell maturation, interferon signalling and NF-kB signalling pathways in circulating T cells and monocytes Patients on DMT showed increased abundances of *Prevotella* and *Sutterella*, and decreased *Sarcina*, compared with untreated patients. MS patients showed elevated breath methane
Cekanaviciute et al. [40]	71		Akkermensia muciniphila and Acinetobacter calcoacetics increased in MS patients Parabacteroides distasonis decreased in MS patients
Berer et al. [52]	68	Comparison between the GM composition of 34 monozygotic twin pairs discordant for MS	No major differences in the overall microbial profiles Significant increase in some taxa such as *Akkermansia* in untreated MS twins
Kouchaki et al. [57]	60	RCT comparing influence of probiotic on MS course	The use of probiotics had favorable effects on EDSS, parameters of mental health, inflammatory factors, markers of insulin resistance
Tankou et al. [54]	9	Investigation of the effect of VSL3 on the gut microbiome and peripheral immune system function in HC and MS	VSL3 administration was associated with increased abundance of many taxa with enriched taxa predominated by *Lactobacillus*, *Streptococcus*, and *Bifidobacterium* species. VSL3 administration induced an anti-inflammatory peripheral immune response
Tankou et al. [56]	9	Administration of probiotic containing *Lactobacillus, Bifidobacterium*, and *Streptococcus*	Probiotic administration increased the abundance of several taxa known to be depleted in MS
Cox et al. [39]	199 RRMS, 44 PMS	Sequencing of microbiota in HC, RRMS and PMS patients, correlation of these with clinical biomarkers	HC and MS differed when it comes to microbiota composition No differences between RRMS and PMS Clostridium species associated with higher EDSS and fatigue scores

MS—multiple sclerosis, GM—gut microbiome, MOG—myelin oligodendrocyte glycoprotein, HC—healthy controls, RRMS—relapsing and remitting multiple sclerosis, PMS—progressive multiple sclerosis, EAE—experimental autoimmune encephalomyelitis, GA—glatiramer acetate, Vit. D—vitamin D, DMT—disease-modifying therapy, BBB—blood-brain barrier, IL—interleukin, RCT—randomized controlled trial, EDSS—Expanded Disability Status Scale, NA – not available

**Table 2 cells-12-01760-t002:** An overview of studies investigating the role of gut microbiome in the pathogenesis of neuromyelitis optica spectrum disorders.

Reference	Number of NMO-SD Patients	Methodology	Results
Varrin-Doyer et al. [66]	15	Peripheral blood T cells were obtained from 15 NMO patients and 9 HC	T cells from NMO patients showed higher proliferation, especially when exposed to the AQP peptide p61–80, cross reactivity to a sequence within *C. perfingens*, and T17 polarization.
Cree et al. [67]	16	Stool samples were collected from 16 NMO patients (all AQP4 seropositive), 16 HC and 16 MS patients with similar nutritional intake	There was a statistically significant difference in the abundance of *C. perfringens* in the stool of NMO patients and HC
Gong et al. [68]	84	Stool samples were obtained from 84 NMO patients and 54 HC.	The *Strepptococcus* sp. were overrepresented in NMO patients, which correlated with the disease severity. There was a significant reductionon of *Faecal butyrate* in NMO patients.
Shi et al. [70]	20	Stool samples were obtained from 20 NMO patients and 20 HC	Overrepresention of the pathogenic species *Streptococcus* and *Flavonifractor* in NMO sample
Cui et al. [69]	6	Sigmoid mucosal biopsies were obtained using endoscopy from 6 NMO patients and 5 HC	The diversy of bacterial flora in NMO patients was diminished. *Streptococcus* and *Granulicatella* sp. were still abundant
Zhang et al. [71]	22	Stool samples were collected from 22 NMO patients (14 AQP seropositive) and 28 HC	AQP4+ and AQP4-groups of NMO patients as well as HC showed the prevalence of different bacteria—*Proteobacteria*, *Bacteroidetes* and *Firmicutes*, respectively. Butyrate-producing species were abundantly represented in HC compared to NMO patients.
Pandit et al. [72]	39	Stool and peripheral blood samples were collected from 39 patients with NMO (17 AQP4 seropositive) and 36 HC.	The prevalence of *C boltae* was significanlty higher in AQP4 seropositive patients. *C boltae* peptide p59–71 showed homology with AQP peptide p 92–104. The presence of *C boltae* correlated with the expression of inflammatory genes.

NMO-SD—neuromyelitis spectrum disorder, AQP—aquaporin, HC—healthy controls.

## Data Availability

Not applicable.

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
