# Peer review of "The Role of Gut Microbiome in the Pathogenesis of Multiple Sclerosis and Related Disorders"

_cells, 2023, doi:10.3390/cells12131760_

Round 1
Reviewer 1 Report
In the manuscript “The Role of Gut Microbiome in the Pathogenesis of Multiple Sclerosis and Related Disorders” the authors summarize some evidence highlighting the role of gut microbiome in the pathophysiology of MS and related disorders, such as neuromyelitis optica spectrum disorder.
The article is well written and well organized. However, some advices could improve the content.
The authors should address these points:
- the authors should specify the literature they have considered, both the source (e.g. Medline, pubmes, Scopus) and the time window.
- the use of the terms microbiome/microbiota could be confusing for the reader. The authors should define both concepts.
- There is a lack of key studies investigating the role of the gut microbiota in MS, both in human samples (e.g. Adv Exp Med Biol. 2016;935:89-98. doi: 10.1007/5584_2016_25. ; https://doi.org/10.3389/fcimb.2016.00198) and in animal models (https://doi.org/10.1080/19490976.2017.1353843.). The authors could consider what has already been summarised ( Int. J. Mol. Sci. 2020, 21, 4045; doi:10.3390/ijms21114045 ).
Tha manuscript is well written.
Author Response
We would like to express our gratitude for the time dedicated to review our manuscript and valuable suggestions. We have revised our manuscript accordingly and provide detailed responses to each point in bold below as well as in track changes mode in the revised version of our manuscript.
In the manuscript “The Role of Gut Microbiome in the Pathogenesis of Multiple Sclerosis and Related Disorders” the authors summarize some evidence highlighting the role of gut microbiome in the pathophysiology of MS and related disorders, such as neuromyelitis optica spectrum disorder.
The article is well written and well organized. However, some advices could improve the content.
The authors should address these points:
- the authors should specify the literature they have considered, both the source (e.g. Medline, pubmes, Scopus) and the time window.
We appreciate this valuable suggestion and have included this information in the new version of the manuscript, both in the abstract as well methodology section: “For this aim, we have conducted systematic review of the literature listed in the following database: Medline, Pubmed and Scopus and were able to identify several studies demonstrating involvement of gut microbiome in pathophysiology of MS and NMO-SD” (lines 18-21) and “To identify articles eligible for inclusion, we have searched electronic databases (Pubmed, Medline and Scopus) with the main focus on original research. We did not use any time restrictions to include the articles” (lines 55-57).
- the use of the terms microbiome/microbiota could be confusing for the reader. The authors should define both concepts.
We agree with this suggestion and, in fact, have included this information in the discussion section: “Another issue is related to the differences between microbiome which constitutes both the host environment as well as all microorganisms encountered in it, and microbiota, which is limited to microorganisms only (70)” (lines 230-232). However, we do agree that it should also be listed in the introduction section since it is confusing for the reader. We have therefore included this sentence: “While microbiome include both the host environment as well as the variety of microorganisms populating it, the term “microbiota” is used to define only microbiota specific for particular host or disease” (lines 42-44).
- There is a lack of key studies investigating the role of the gut microbiota in MS, both in human samples (e.g. Adv Exp Med Biol. 2016;935:89-98. doi: 10.1007/5584_2016_25. ; https://doi.org/10.3389/fcimb.2016.00198) and in animal models (https://doi.org/10.1080/19490976.2017.1353843.). The authors could consider what has already been summarised ( Int. J. Mol. Sci. 2020, 21, 4045; doi:10.3390/ijms21114045 ).
Following this suggestion, we have included the missing studies in the amended version of the manuscript. We have also alluded to aforementioned review and have summarized missing studies in the new version of the paper.
Reviewer 2 Report
The current article summarizes current evidence confirming the role of the gut microbiome in the pathophysiology of MS and related disorders. I have two main suggestions:
1- please add a paragraph discussing the methods used for literature research and selection;
2- Please try to improve the discussion on the potential use of therapies that interfere with the composition of the gut microbiome (probiotics and fecal transplantation) for MS. Is there any study already available?
Only minor editing required
Author Response
We would like to express our gratitude for the time dedicated to review our manuscript and valuable suggestions. We have revised our manuscript accordingly and provide detailed responses to each point in bold below as well as in track changes mode in the revised version of our manuscript.
The current article summarizes current evidence confirming the role of the gut microbiome in the pathophysiology of MS and related disorders. I have two main suggestions:
- please add a paragraph discussing the methods used for literature research and selection;
We appreciate this valuable suggestion and have included this information in the new version of the manuscript, both in the abstract as well methodology section: “For this aim, we have conducted systematic review of the literature listed in the following database: Medline, Pubmed and Scopus and were able to identify several studies demonstrating involvement of gut microbiome in pathophysiology of MS and NMO-SD” (lines 18-21) and “To identify articles eligible for inclusion, we have searched electronic databases (Pubmed, Medline and Scopus) with the main focus on original research. We did not use any time restrictions to include the articles” (lines 55-57).
- Please try to improve the discussion on the potential use of therapies that interfere with the composition of the gut microbiome (probiotics and fecal transplantation) for MS. Is there any study already available?
We also agree with this recommendation and have included more information about this topic: “Finally, it is also still not clear whether active interventions changing micriobiome composition, such as probiotics supplementation (96)or fecal transplantation (97). In the recently published systematic review and meta-analysis by Jiang et al. (96)the authors summarized previous evidence behind the use of probiotics as treatment of MS and were able to identify 3 RCTs and 22 preclinical studies. Meta-analysis of RCTs demonstrated significant benefit of probiotic supplementation on mental health , but the certainty of evidence was very low. Moreover, probiotic administration lead to improvement of insulin resistance, inflammatory and oxidative stress markers. Preclinical studies showed that probiotics reduce the incidence and severity of MS, delays MS progression and improves motor impairment. All in all, these results indicate that probiotics have beneficial effect on disease progression in MS and constitute another argument supporting microbiome importance in MS development. Fecal transplantation was only tested in very small number of patients and have shown improvement of intestinal permeability, but results regarding symptom improvement could only been tested in bigger samples (97–99). ” (lines 281-294).
We also attach our revised paper in the track change mode.
Reviewer 3 Report
The review is entitled “The Role of Gut Microbiome in the Pathogenesis of Multiple Sclerosis and Related Disorders”. The review is exciting but has several significant issues that must be addressed. Here are my comments:
1. In the abstract section, the authors should write the background of the review. Why is this review important?
2. Authors should provide more literature; I suggest an elaborate authors introduction. The introduction needs revision based on the updated literature.
3. Explain the experimental animal models of MS? and the role of the Gut Microbiome in the MS mouse model.
4. Authors should include a table/cartoon showing the pathways that could be targeted in the Gut Microbiome in the Pathogenesis of MS. Diagrammatic representations are significant in a review. You should have at least two diagrammatic representations.
5. The limitations of the review should be shared in the discussion forum, clarifying the scientific research community.
6. The authors should carefully re-organize the structure errors and improve the language.
Author Response
We would like to express our gratitude for the time dedicated to review our manuscript and valuable suggestions. We have revised our manuscript accordingly and provide detailed responses to each point in bold below as well as in track changes mode in the revised version of our manuscript.
The review is entitled “The Role of Gut Microbiome in the Pathogenesis of Multiple Sclerosis and Related Disorders”. The review is exciting but has several significant issues that must be addressed. Here are my comments:
- In the abstract section, the authors should write the background of the review. Why is this review important?
Thank you for this suggestion. We agree that it is an important point and have included it in the new version of the paper: “The topic gut microbiome in pathophysiology of MS is therefore relevant since it could be used as biomarker of disease development and progression as well as disease-modifying therapy” (lines 27-29).
- Authors should provide more literature; I suggest an elaborate authors introduction.The introduction needs revision based on the updated literature.
We also agree with this comment and have updated our review with more literature and have also expanded the introduction that could be consulted in the revised version of the manuscript in the track changes mode (lines 33-79).
- Explain the experimental animal models of MS? and the role of the Gut Microbiome in the MS mouse model.
We agree that it is important to explain with more detail what animal models of MS exist since there is a lot of evidence supporting the role of gut microbiome in MS deriving from animal models: “Animal models provide information that MS-linked microbiota produce factors that precipitate an MS-like autoimmune disease (36). Molecular mimicry between gut bacterial component and central nervous system (CNS) autoantigen, in concert with gut microbes inducing Th17 cells, act together to worsen CNS autoimmunity in an experimental autoimmune encephalomyelitis (EAE), most widely used animal model of MS (37). Therefore, the main mechanism behind the creation of this animal model, is based on gut-brain interaction. In the process of development of this model, it has been found that antibiotic administration protected against the disease, which was attributed to Treg- and Th2-cell responses (38). (lines 84-94).
- Authors should include a table/cartoon showing the pathways that could be targeted in the Gut Microbiome in the Pathogenesis of MS.Diagrammatic representations are significant in a review. You should have at least two diagrammatic representations.
We appreciate this suggestion and agree that figures are very illustrative and help to conceptualize better the ideas of the authors and summarize previous evidence. We have included figure summarizing the influence of environmental factors on gut microbiome.
- The limitations of the review should be shared in the discussion forum, clarifying the scientific research community.
We also appreciate this suggestion and have included appropriate paragraph in the discussion section: There are several limitations of our review: i) although we used detailed search strategy, it is still possible that we did not include some important studies, especially the ones posterior to publication of our review; ii) another limitation is related to the inclusion of articles that are written only in English; iii) finally, we mainly focused on the microbiome in our review, but it is highly dependent on environmental factors(96), especially diet that also plays an important role in development of MS(97) (lines 279-284).
- The authors should carefully re-organize the structure errors and improve the language.
Thank you for this suggestion. We have revised the manuscript for language and structural errors.
Round 2
Reviewer 2 Report
Accept
Reviewer 3 Report
The authors have answered my questions, and the paper has significantly improved.